# Efficacy of Oral Supplementation with Cholecalciferol Versus Calcifediol in Patients with Hypovitaminosis D After Stroke

**DOI:** 10.3390/nu17061035

**Published:** 2025-03-15

**Authors:** Stefania Canneva, Anna De Giovanni, Felicita Pagella, Lucia Pellegrino, Francesco Iencinella, Sara Maestrini, Marta Ponzano, Carlo Trompetto, Laura Mori

**Affiliations:** 1IRCCS Ospedale Policlinico San Martino, Largo Rosanna Benzi 10, 16132 Genoa, Italy; stefania.canneva@hsanmartino.it (S.C.); felicitapagella@gmail.com (F.P.); pellegrinolucia.med@gmail.com (L.P.); francescoiencinella.mmg@gmail.com (F.I.); sara.maestrini91@gmail.com (S.M.); ctrompetto@neurologia.unige.it (C.T.); 2Department of Neurosciences, Rehabilitation, Ophthalmology, Genetics and Maternal and Child Sciences (DINOGMI), University of Genoa, 16126 Genova, Italy; 3Department of Health Sciences (DISSAL), University of Genoa, 16126 Genova, Italy; ponzano.marta@gmail.com

**Keywords:** vitamin D deficiency 1, cholecalciferol 2, calcifediol 3, stroke 4, rehabilitation 5, osteoporosis 6

## Abstract

Background/Objectives: Cholecalciferol and Calcifediol are commonly used for oral supplementation in patients with vitamin D deficiency. Several studies have compared these two molecules; however, the studied population has only included healthy postmenopausal women so far. This retrospective observational study aims to evaluate which molecule is more effective and faster in achieving serum 25(oh)D levels within the normal range in post-stroke patients during the subacute phase. Secondary aims include assessing potential differences in functional outcomes and investigating the possible correlation between the degree of hypovitaminosis D and stroke severity. Methods: We observed 85 in-patients who received either Cholecalciferol or Calcifediol during intensive rehabilitation. All subjects underwent functional evaluations, blood tests, and a bone densitometry (DEXA) scan. Results: Four months after starting supplementation, subjects receiving calcifediol achieved significantly higher 25(oh)D levels (*p* < 0.001) compared to those receiving cholecalciferol. No significant between-group differences were observed in secondary outcomes. Another key finding is that no statistically significant correlation was found between serum of 25(oh)D levels and stroke severity. Conclusions: These results highlight the importance of further investigating bone metabolism in post-stroke patients, though findings should be confirmed in larger studies.

## 1. Introduction

Vitamin D is essential for bone metabolism as it induces calcium and phosphorus gastrointestinal absorption and reduces mRNA-parathyroid hormone expression in parathyroids. These actions help maintain a correct serum calcium and phosphorus balance, positively impacting bone health. Extra-skeletal effects have also been documented in several systems, including the immune and cardiovascular systems (regulation of renin and insulin synthesis), and in preventing the development of neoplasia by controlling cellular proliferation and differentiation [1]. Moreover, as a neurosteroid, vitamin D influences CNS development and brain functions, playing a role in neuroplasticity and preventing axonal degeneration due to its widespread receptor distribution on neural and glial cells [2,3]. Hypovitaminosis D represents a public health concern, being an independent risk factor for mortality in the general population [4]. Low 25(oh)D serum levels are a negative prognostic factor in patients with multiple sclerosis, Parkinson’s disease, amyotrophic lateral sclerosis, and stroke [5]. In Italy, stroke is the leading cause of disability and the third leading cause of death, following heart disease and cancer. Studies suggest that maintaining appropriate serum 25(oh)D levels is associated with improved cardiovascular function and a consequent reduction in stroke incidence [2]. Additionally, research has investigated the correlation between vitamin D and stroke pathophysiology [6,7,8]. Specifically, vitamin D deficiency is associated with increased inflammatory mediators such as Interleukin 6 [9]. Furthermore, vitamin D promotes the expression of IGF1, a neuroprotective hormone with antithrombotic action that prevents axonal and dendritic degeneration [10,11,12], and nitric oxide, a potent vasodilator involved in neuronal survival mechanisms [10,11]. Several studies have reported an association between serum 25(oh)D levels and functional recovery in post-stroke patients, highlighting that low 25(oh)D levels correlate with worse functional outcomes [9,13]. Stroke patients often present with a general malnutrition profile that can impair functional recovery. Therefore, careful monitoring of nutritional indices, specifically serum 25(oh)D levels, is recommended due to its beneficial effects not only on bone health but also on balance and the muscular system. Given the high prevalence of hypovitaminosis D and its correlation with functional prognosis, vitamin D supplementation should always be considered in patients with a history of stroke [6]. According to guidelines, levels above 30 ng/mL are considered optimal; lower values are deemed insufficient, necessitating adequate supplementation. Cholecalciferol and calcifediol are two available molecules for hypovitaminosis D treatment. They differ in terms of half-life (25–30 days for cholecalciferol versus 10–15 days for calcifediol) and potency (calcifediol appears to be three times more effective than cholecalciferol). In clinical practice, cholecalciferol is the most commonly used, being the reference molecule for osteoporosis patients due to its anti-fracture efficacy. Calcifediol, being the circulating form, is preferred in individuals with hepatic storage and metabolism alterations (e.g., obesity, excessive thinness, elderly subjects, hepatic insufficiency, and malabsorption). Additionally, it seems to have more positive effects on immune system regulation and arterial blood pressure compared to cholecalciferol [14]. Although many studies have compared these two molecules, the reference population has only included healthy postmenopausal women thus far [15,16,17,18,19,20,21].

Our study aims to determine which molecule is more potent and rapid in achieving normal serum 25(oh)D levels in post-stroke patients during the subacute phase. Secondary aims include evaluating potential differences in improving functional outcomes and verifying the correlation between the degree of hypovitaminosis D and stroke severity.

## 2. Materials and Methods

In this retrospective study, we observed 85 in-patients (mean age 66 ± 11 years), 51 males (60%) and 34 females (40%), admitted to the Neurorehabilitation unit of IRCCS Ospedale Policlinico San Martino of Genova, Italy, between June 2022 and June 2024. Sixty-seven (79%) of the included subjects had a diagnosis of ischemic stroke and eighteen (21%) of hemorrhagic stroke. The inclusion criteria were a diagnosis of subacute stroke with intact cognitive function, hypovitaminosis D (25(oh)D serum levels < 30 ng/mL) supplemented with Cholecalciferol or Calcifediol. Specifically, a dose of 20 mcg per day of calcifediol was administered, while as for cholecalciferol, the dosage was adjusted according to serum levels of 25(oh)D, using loading doses of 100,000 IU or 300,000 IU (divided into weekly doses of 25,000 IU) in cases of severe hypovitaminosis D (300,000 IU if 25(oh)D < 10 ng/mL, 100,000 IU if 25(oh)D < 20 ng/mL) in accordance with good clinical practice guidelines [22]. Subjects with psychiatric conditions, blindness, or severe aphasia were excluded. Upon admission to the Neurorehabilitation unit (T0), the following evaluation scales were administered according to clinical practices: National Institute of Health Stroke Scale (NIHSS) for stroke severity, Modified Barthel Index (BIM) for daily life activities, Short Physical Performance Battery (SPPB), and Visual Analogue Scale (VAS) for pain perception. A quality-of-life questionnaire (SF12) was also administered, and premorbid disability was assessed using the Modified Rankin Scale (MRS). Additionally, blood tests were performed to assess bone turnover markers (25(oh)D, parathyroid hormone, calcium, phosphorus), and bone density was evaluated using a DEXA scan. All subjects underwent a one-month tailored intensive rehabilitation program.

All patients were assessed upon admission to the rehabilitation unit (T0), and 53 patients (62%) were re-evaluated after four months (T1) during the follow-up visit. Assessments included blood tests for bone metabolism and the following evaluation scales: VAS, SPPB, and SF-12.

For statistical analysis, the Student’s *t*-test for unpaired data and the Mann–Whitney test for continuous variables, as well as the chi-square test for categorical variables, were used. Correlations were investigated using Pearson’s test. A two-tailed α level of less than 0.05 was considered statistically significant. Our study was approved by the Territorial Ethics Committee (N. CET—Liguria: 384/2024-DB id 14133).

## 3. Results

The two groups of patients were homogeneous at T0 across all examined variables (Table 1).

At T1, all treated patients showed a statistically significant improvement (*p* < 0.001) in 25(oh)D blood levels, regardless of the type of vitamin supplement used (see Figure 1A). However, patients treated with calcifediol (cal) achieved higher serum levels of 25(oh)D at T1 compared to those treated with cholecalciferol (chol), with a statistically significant difference (*p* < 0.001) (see Figure 1B).

Serum calcium, phosphorus, and parathyroid hormone levels remained within normal ranges in both groups, indicating a favorable safety profile for both molecules.

In all patients assessed at T1, a statistically significant improvement was observed in motor performance (SPPB scale) and subjective perception of their functional abilities (SF12 physical composite score—SF12 PCS) (*p* < 0.001 and *p* = 0.002) (see Figure 2A,B).

No statistically significant correlation was found between serum levels of 25(oh)D and the severity of stroke, assessed using the NIHSS scale (rho = −0.14, *p* = 0.21) (see Figure 3). A statistically significant moderate negative correlation (rho = −0.56; *p* < 0.001) was found between stroke severity (NIHSS scale) and associated disability (BIM) (see Figure 3).

## 4. Discussion

In clinical practice, both cholecalciferol and calcifediol are used without specific guidelines on which molecule is preferable for patients with post-stroke outcomes. This is the first study in the literature aimed at demonstrating which of the two supplements (cholecalciferol versus calcifediol) is more effective in restoring serum 25(oh)D levels to the reference range in patients with subacute post-stroke outcomes. Simultaneously, we aimed to identify statistically significant differences in terms of motor performance, pain perception, or quality of life. Unlike previous studies in the literature [9,13,14], the patients in our study who were most neurologically impaired at the time of admission to our Neurorehabilitation department did not present with severe 25(oh)D deficiency. Moreover, no statistically significant correlation was found between serum levels of 25(oh)D and the severity of stroke at the NIHSS scale. This evidence may be related to the different sample numerosity compared to the studies of Alharbi and other authors [8,9,10,13,23].

All patients who continued regular supplementation reported no side effects during intake. Four months after the start of supplementation, serum 25(oh)D levels increased significantly in both groups, but higher levels were achieved in patients treated with calcifediol compared to those treated with cholecalciferol. This finding is consistent with the literature, which sampled healthy postmenopausal women. However, this did not correspond to an improvement in motor performance, pain perception, or quality of life. Calcium, phosphorus, and parathyroid hormone levels remained within normal ranges in both groups, indicating a good safety profile for both molecules. The SPPB and the SF12-PCS showed statistically significant improvements in all patients from T0 to T1, highlighting the importance of intensive physiotherapy in the post-acute phase of stroke alongside intrinsic mechanisms of cerebral plasticity. Of all the patients, two initiated antiresorptive therapy with bisphosphonates at T0 along with vitamin D supplementation due to a diagnosis of osteoporosis (T-score < 3 on MOC-DEXA) in association with risk factors for bone fragility. This finding is crucial for patients with subacute post-stroke outcomes, where proper evaluation of bone metabolism is necessary. It has been demonstrated that osteoporosis predominates on the paretic side [24], and within the first year, patients can lose more than 14% of their body mass index at the proximal femur [25,26] and more than 17% at the distal upper limb [26,27,28].

Hypovitaminosis D plays a crucial role in both the functionality and pathology of the nervous system. However, the literature contains numerous studies with controversial results regarding the correlation between hypovitaminosis D and the incidence of neurological diseases. For example, a 2013 review [5] indicated that hypovitaminosis D leads to greater susceptibility to various psychiatric and neurological disorders (e.g., schizophrenia, autism, Parkinson’s disease, amyotrophic lateral sclerosis, Alzheimer’s disease, multiple sclerosis). Additionally, numerous scientific studies attempt to correlate blood 25(oh)D concentrations with the onset and progression of various neurological diseases, such as multiple sclerosis, Parkinson’s disease, Alzheimer’s disease, migraine, diabetic neuropathy, and amyotrophic lateral sclerosis [29,30,31].

The literature also highlights that 25(oh)D concentrations above 20 ng/mL are associated with a significant reduction in the risk of stroke and cardiovascular events, seemingly in a random manner; therefore, achieving levels above 20 ng/mL might be recommended in the general population, particularly for individuals at risk of cardio- and cerebrovascular events [32]. However, there are currently no definitive guidelines regarding the efficacy of vitamin D supplementation in the etiopathogenesis and progression of nervous system disorders. Nonetheless, the scientific literature agrees with considering serum 25(oh)D levels as a marker of good general health and an adequate lifestyle [29]. In fact, low 25(oh)D levels may reflect dietary habits and levels of physical activity and can be influenced by many health factors, including body mass index [30]. A recent 2022 study by Abdullah R. Alharbi et al. highlighted that low blood levels of 25(oh)D, measured within 7 days of symptom onset in patients with cerebral ischemia, are associated with greater stroke severity at admission and poor functional recovery at discharge [33]. However, it remains unclear whether low 25(oh)D levels contribute to worse outcomes or are instead a consequence of the ischemic event itself, possibly due to metabolic stress and systemic inflammation [34]. Various studies in the literature have investigated different dosages and duration of vitamin D supplementation. Narasimhan et al.’s 2017 study [35] found that a single intramuscular dose of 600,000 IU cholecalciferol led to improvement on the Scandinavian Stroke Scale 12 weeks after stroke. In contrast, Gupta et al.’s 2016 study [36] noted that a single intramuscular dose of 600,000 IU cholecalciferol followed by 60,000 IU oral cholecalciferol for one month increased the likelihood of survival 24 weeks post-stroke. A recent 2021 retrospective study by Karasu et al. divided patients with ischemic/hemorrhagic stroke outcomes undergoing intensive rehabilitation into two groups. It demonstrated that the group treated with cholecalciferol (50,000 IU per week for 4–12 weeks) had a statistically significant improvement in the Functional Assessment of Movement (FAC) and in the Brunnstrom Recovery Stage (BRS) compared to the placebo group [37]. Conversely, Momosaki’s 2019 study suggested that daily administration of 2000 IU of cholecalciferol did not lead to better functional outcomes compared to the placebo group [38]. Similarly, Torrisi, Bonanno, Formica et al.’s 2021 study on post-stroke patients undergoing intensive rehabilitation found that the group treated with 2000 IU daily cholecalciferol did not show greater functional improvements compared to the placebo group [39]. The 2021 study by Sari, Durman, Karaman, et al. evaluated the effects of cholecalciferol supplementation in patients with hemiplegia due to ischemic stroke. The study concluded that, compared to the placebo group, patients who received a single intramuscular dose of 300,000 IU cholecalciferol showed significant improvement in basic activities of daily living, as measured by the BIM, but not in movement or motor recovery [40]. In general, considering the potential beneficial effects and limited associated risks, the literature supports the initiation of appropriate vitamin D supplementation based on observed serum levels and the evaluation of neurological patients from the perspective of bone metabolism. This assessment is crucial to prevent a reduction in bone mineral density, which can lead to fragility fractures. Such an event, especially in “fragile” patients, causes a global functional decline and increased mortality risk. It is well established that hypovitaminosis D represents a public health issue, being an independent risk factor for mortality in the general population [4].

However, Bouillon et al. state that many individuals take vitamin D supplementation without a clear clinical benefit, and a small percentage of the general population takes higher doses than the phase intake; therefore, the authors conclude that vitamin D supplementation should be administered only when necessary, based on clinical-anamnestic data [41].

The main limitation of our study is the small sample size. Additionally, it is a retrospective and observational study; therefore, a future randomized controlled trial will be necessary to validate the findings. Another limitation is the absence of a control group, which would allow verification of the actual effects of dietary vitamin D supplementation and its potential correlation with outcome measures.

A general limitation of prospective studies is also the tendency to underestimate the association between disease and risk factors due to the “regression dilution ratio” phenomenon [42]. In the specifics of our work, therefore, 25(oh)D levels may fluctuate over time, and a single measurement at baseline may not accurately represent long-term exposure. This could lead to an underestimation of the association between 25(oh)D and outcomes in post-stroke patients. Future research should focus on conducting further studies to elucidate the role of vitamin D in patients with chronic post-stroke outcomes, assessing the effect of the supplementation on bone density and fall risk without interference with motor recovery. Future studies should also consider dietary and lifestyle factors (nutrition, physical activity, and sun exposure) that influence vitamin D metabolism, even though oral supplementation is recognized as the most effective approach [43]

## 5. Conclusions

Our study demonstrated that both calcifediol and cholecalciferol are safe and effective molecules in restoring 25(oh)D levels to the reference range, with calcifediol exhibiting a more rapid action. However, no significant differences were observed in terms of functional outcomes between the two compounds when considering their efficacy. Another important finding is that no statistically significant correlation was found between serum levels of 25(oh)D and the severity of stroke. These data suggest the importance of continuously studying bone metabolism in these patients, although these findings should be verified in a larger sample size.

## Figures and Tables

**Figure 1 nutrients-17-01035-f001:**
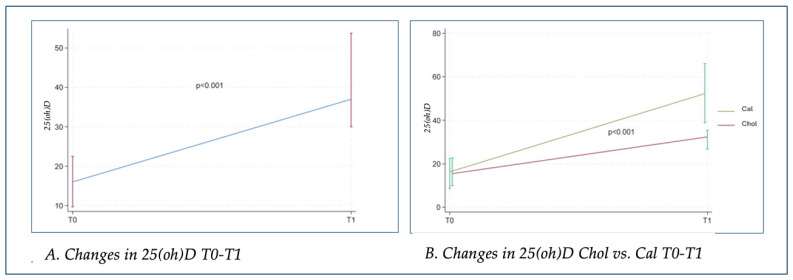
Changes in 25(oh)D: (**A**) changes in all subjects; (**B**) changes in subjects treated with cholecalciferol or calcifediol. 25(oh)D: 25OH vitamin D; Chol: cholecalciferol; Cal: calcifediol.

**Figure 2 nutrients-17-01035-f002:**
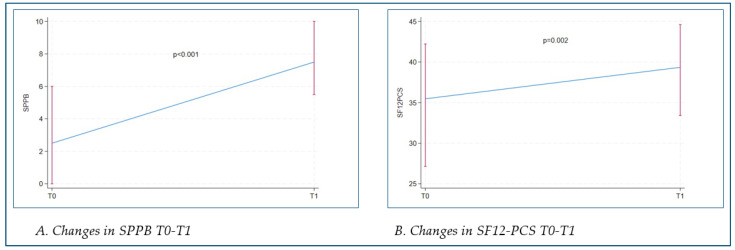
(**A**) Changes in SPPB or all subjects; (**B**) changes in SF12-PCS in all subjects. SPPB: Short Physical Performance Battery; SF12: short form 12; PCS: physical composite score.

**Figure 3 nutrients-17-01035-f003:**
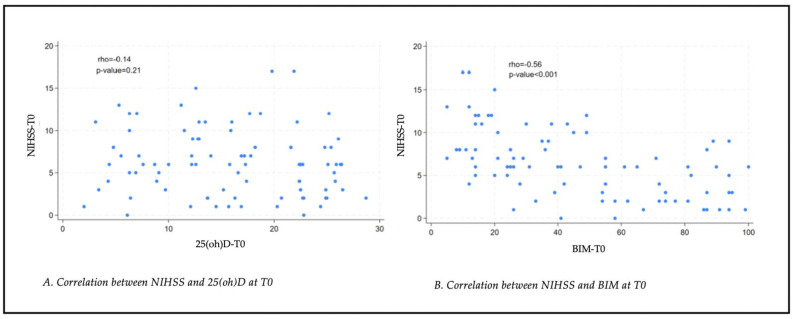
(**A**) Correlation between NIHSS and 25(oh)D at T0; (**B**) correlation between NIHSS and BIM at T0. NIHSS: National Institutes of Health Stroke Scale; 25(oh)D: 25Oh vitamin D; BIM: Modified Barthel Index.

**Table 1 nutrients-17-01035-t001:** Summary table of variables at T0.

	Calcifediol (N = 43)	Cholecalciferol (N = 42)	*p*-Value
Age, median (IQR)	67 (61; 76)	66.5 (59; 73)	0.46
Female, N (%)	19 (44%)	15 (36%)	0.45
Ischemic stroke, N (%)	34 (79%)	33 (79%)	0.96
NIHSS T0, median (IQR)	6 (3; 9)	6 (3; 9)	0.57
BIM T0, median (IQR)	40 (21; 81)	41 (20; 65)	0.61
SPPB T0, median (IQR)	0 (0; 7)	3 (0; 6) (N = 41)	0.57
SF12PCS T0, median (IQR)	35.9 (26.24; 42.05) (N = 42)	34.64 (31.54; 42.9) (N = 41)	0.59
SF12MCS T0, median (IQR)	46.12 (34.71; 53.1) (N = 42)	46.5 (35.29; 56.92) (N = 41)	0.56
VAS, median (IQR)	0 (0; 6)	0 (0; 4) (N = 41)	0.50
25(oh)D T0, median (IQR)	16.3 (8.7; 22.4)	15.5 (10; 22.7)	0.93
Calcium T0, median (IQR)	9.3 (9.1; 9.5)	9.45 (9.1; 9.8)	0.10
Phosphorus T0, median (IQR)	3.3 (3; 3.7) (N = 42)	3.2 (2.9; 3.6)	0.42
PTH T0, median (IQR)	45.5 (29; 66) (N = 42)	35.5 (27; 51) (N = 36)	0.21
T-SCORE SPINE T0, median (IQR)	−1 (−2.1; 0.6) (N = 39)	−0.1 (−1.6; 0.9) (N = 39)	0.32
T-SCORE NECK T0, median (IQR)	−1.5 (−2.5; −0.8) (N = 39)	−1.3 (−1.7; −0.8) (N = 40)	0.46
T-SCORE INTER T0, median (IQR)	−1.1 (−1.9; 0.4) (N = 39)	−0.75 (−1.3; 0.1) (N = 40)	0.39

NIHSS: National Institute of Health Stroke Scale; BIM: Modified Barthel Index; SPPB: Short Physical Performance Battery; SF12: Short form 12; PCS: physical composite score; MCS: mental composite score; VAS: visual analogic scale for pain; 25(oh)D: 25OH vitamin D; PTH: parathyroid hormone.

## Data Availability

The data supporting the reported results are not publicly available due to privacy and ethical restrictions and cannot be shared. The data presented in this study are available on request from the corresponding author upon reasonable request, in compliance with ethical and legal regulations.

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
