# Peer review of "Efficacy of Oral Supplementation with Cholecalciferol Versus Calcifediol in Patients with Hypovitaminosis D After Stroke"

_nutrients, 2025, doi:10.3390/nu17061035_

Round 1

Reviewer 1 Report

Comments and Suggestions for Authors

The role of follow-up period in prospective cohort studies regarding findings from prospective cohort studies should be discussed:

Underestimation of risk associations due to regression dilution in long-term follow-up of prospective studies. Clarke R, Shipley M, Lewington S, Youngman L, Collins R, Marmot M, Peto R. Am J Epidemiol. 1999 Aug 15;150(4):341-53. doi: 10.1093/oxfordjournals.aje.a010013.

How Follow-Up Period in Prospective Cohort Studies Affects Relationship Between Baseline Serum 25(OH)D Concentration and Risk of Stroke and Major Cardiovascular Events. Grant WB, Boucher BJ. Nutrients. 2024 Nov 1;16(21):3759. doi: 10.3390/nu16213759.

Follow-Up Period Affects the Association between Serum 25-Hydroxyvitamin D Concentration and Incidence of Dementia, Alzheimer's Disease, and Cognitive Impairment. Grant WB. Nutrients. 2024 Sep 23;16(18):3211. doi: 10.3390/nu16183211.

Four months after starting supplementation, subjects receiving calcifediol achieved

22

significantly higher 25OH vitamin D levels (P<0.001) compared to those receiving

23

cholecalciferol.

Comment: This finding does not demonstrate that calcifediol is better than cholecalciferol in improving 25(OH)D over four months. Calcifediol raises 25(OH)D faster than cholecalciferol, but a bolus dose of cholecalciferol would do about the same. Had a higher dose of cholecalciferol been used, the finding could have been reversed. It should also be noted that calcifediol is not readily available in many countries including the U.S. As for Spain, it is my understanding that since both forms are sold by prescription, the prices are similar.

This paper should be discussed: Human serum 25-hydroxycholecalciferol response to extended oral dosing with cholecalciferol. Heaney RP, Davies KM, Chen TC, Holick MF, Barger-Lux MJ.Am J Clin Nutr. 2003 Jan;77(1):204-10. doi: 10.1093/ajcn/77.1.204.

Table 1 should define DY, DB in the description or a footnote.

A recent 2022 study by Abdullah R. Alharbi et al. 206 highlighted that low blood levels of 25-hydroxyvitamin D3 measured within 7 days of 207 symptom onset in patients with cerebral ischemia are associated with greater stroke 208 severity at admission and poor functional recovery at discharge [29].

Comment: It is likely that having a stroke reduced 25(OH)D. See, e.g.,

Letter to the Editor: Vitamin D deficiency in COVID-19: Mixing up cause and consequence. Smolders J, van den Ouweland J, Geven C, Pickkers P, Kox M.Metabolism. 2021 Feb;115:154434. doi: 10.1016/j.metabol.2020.154434.

Low vitamin D levels

25OH 127 vitamin D3 blood levels,

serum 25-hydroxyvitamin 162 D3 levels

Comment: Please harmonize such statements.

25(OH)D is more accurate as is serum.

Significant digits. The general rule is that no more non-zero digits should be given than are justified by the uncertainty of the value.

See "Too many digits: the presentation of numerical data"

https://www.ncbi.nlm.nih.gov/pmc/articles/PMC4483789/                              

If the uncertainty (or difference when comparing numbers) is greater than about 7%, only two non-zero digits are justified.

P values should be given to two decimal places unless the first two are 00 or the number lies between 0.045 and 0.054. If the first two are 00, then only one non-zero digit can be given.

Thus, p values should be revised.

Percentages should be given in whole numbers due to the small N.

Please review all numbers in abstract, text, tables, and figures and adjust accordingly.

Comments on the Quality of English Language

The English is OK

Author Response

Thank you very much for taking the time to review this manuscript. Please find the detailed responses below and the corresponding revisions highlighted in the re-submitted files

We sincerely appreciate the reviewers' insightful comments and suggestions, which have significantly contributed to refining our work and enhancing the quality of its content.

We discussed the impact of the follow-up period in prospective cohort studies, particularly the underestimation of risk associations due to regression dilution in long-term follow-up (see the Discussion section, lines 256–261).

We acknowledge the reviewer's comment regarding the differences between calcifediol and cholecalciferol in raising 25(OH)D blood levels. These aspects have been addressed in the Materials and Methods section (see lines 93–97).

We have revised the text to clarify the potential bidirectional relationship between low 25(OH)D levels and stroke severity. Specifically, we have acknowledged that it remains unclear whether low 25(OH)D levels contribute to worse outcomes or are a consequence of the ischemic event itself, possibly due to metabolic stress and systemic inflammation (see the Discussion section, lines 211–216)

We have revised the acronyms in Table 1 to improve readability for the reader. Additionally, as recommended, we have adjusted the presentation of numerical data and percentages.  We also have  harmonized the terminology throughout the manuscript, consistently using 'serum 25(OH)D' for accuracy and clarity. These modifications have significantly enhanced the clarity and comprehensibility of the text.

Reviewer 2 Report

Comments and Suggestions for Authors

An interesting study. You looked at functional outcomes as well as biochemical results (25OH vitamin D levels), which are more important for patient recovery in a rehabilitation context.

As demonstrated in this study, calcifediol raises serum 25OH vitamin D levels more effectively than cholecalciferol, as has been repeatedly demonstrated in studies. A review by Bouillon et al. (2019) revealed that long-term gains on functional outcomes might not differ significantly between the two, but previous research has also shown that this does not always transfer to improved clinical advantages in terms of bone health or muscle function (Endocrine Reviews, Volume 40, Issue 4, August 2019, Pages 1109–1151, https://doi.org/10.1210/er.2018-00126).

When it is a retrospective observational study, selection bias may be introduced when randomization is not used. The results may have been impacted by uncontrolled confounding variables, such as comorbidities, seasonal change, sun exposure, and food intake.

You compare two types of vitamin D in your study, but you didn't include a control group that didn't take supplements to find out how rehabilitation alone affected functional outcomes and vitamin D level at baseline.

You claim that the functional outcomes did not differ significantly. Because of this, determining the findings' clinical significance is challenging.

When it comes to vitamin D intake, food sources (such as fatty fish and fortified dairy products) typically offer lower quantities than supplements. Furthermore, cholecalciferol, which makes up the majority of dietary vitamin D, has a slower absorption than calcifediol. According to certain research, vitamin D supplements can quickly address deficiencies more effectively than dietary consumption alone.

Suggestions for Upcoming Studies
o RCT Design: A randomized controlled trial (RCT) is required to validate the findings because this investigation is retrospective and observational.

o Long-Term Effects: More insightful findings may be obtained by evaluating the long-term effects of vitamin D supplementation on bone density, fall risk, and stroke recovery.

o Dietary and Lifestyle Factors: Future research should take into account factors such as nutrition, physical exercise, and sun exposure to help account for outside influences on vitamin D metabolism.

Author Response

Thank you very much for taking the time to review this manuscript. Please find the detailed responses below and the corresponding revisions highlighted in the re-submitted files.

We sincerely appreciate the reviewers' insightful comments and suggestions, which have significantly contributed to refining our work and enhancing the quality of its content.

In response to the reviewer’s comment, we have incorporated relevant points from the Bouillon et al. (2019) review into the manuscript. Specifically, we have added the statement that many individuals take vitamin D supplementation without clear clinical benefit, and that a small percentage of the general population consumes higher doses than the recommended intake. Bouillon et al. conclude that vitamin D supplementation should be administered only when necessary, based on clinical and anamnestic data (see the Discussion section, lines 246–250).

We also appreciate the reviewer’s comment regarding the potential limitations of our study. As noted in the revised manuscript, we have acknowledged that the retrospective and observational nature of our study introduces the possibility of selection bias and uncontrolled confounding variables, such as comorbidities, seasonal changes, sun exposure, and dietary intake. We have also recognized that the absence of a control group limits our ability to assess the true effects of vitamin D supplementation on functional outcomes and its potential correlation with baseline vitamin D levels. These limitations are now highlighted in the Discussion section, where we emphasize the need for future randomized controlled trials to validate our findings (see the Discussion section, lines 251–255 and 261–267).

Round 2

Reviewer 1 Report

Comments and Suggestions for Authors

Vitamin D is essential for bone metabolism as it induces calcium and phosphorus 39 gastrointestinal absorption and reduces mRNA-parathyroid hormone expression in 40 parathyroids. These actions help maintain a correct serum calcium and phosphorus 41 balance, positively impacting bone health. Extra-skeletal effects have also been 42 documented in several systems, including the immune and cardiovascular systems 43 (regulation of renin and insulin synthesis), and in preventing the development of neoplasia 44 by controlling cellular proliferation and differentiation [1].

[1] R Clarke, M Shipley, S Lewington, L Youngman, R Collins, M Marmot, R Peto Underestimation of risk associations 298 due to regression dilution in long-term follow-up of prospective studies. PMID: 10453810 DOI: 10.1093/oxfordjour- 299 nals.aje.a010013

Comment: Clarke et al. is not the appropriate reference for this statement.

Shouldn’t this review be cited?

Vitamin D3 and ischemic stroke: a narrative review W LasoÅ„, D Jantas, M LeÅ›kiewicz, M Regulska… - Antioxidants, 2022 - mdpi.com

And this one?

Combined 25-hydroxyvitamin D concentrations and physical activity on mortality in US stroke survivors: findings from the NHANES J Liao, J Chen, H Wu, Q Zhu, X Tang, L Li, A Zhang… - Nutrition Journal, 2025 - Springer

Significant digits. The general rule is that no more non-zero digits should be given than are justified by the uncertainty of the value.

See "Too many digits: the presentation of numerical data"

https://www.ncbi.nlm.nih.gov/pmc/articles/PMC4483789/                                

If the uncertainty (or difference when comparing numbers) is greater than about 7%, only two non-zero digits are justified.

P values should be given to two decimal places unless the first two are 00 or the number lies between 0.045 and 0.054. If the first two are 00, then only one non-zero digit can be given.                                        

Thus, p values should be revised.

Please review all numbers in abstract, text, tables, and figures and adjust accordingly.

Author Response

We thank the reviewer for identifying the erroneous inclusion of citation 1. We have revised the citation and replaced it with the correct one.

We have corrected the significant digits in the table and in the text.

We sincerely appreciate the reviewer's insightful comments, which have significantly enhanced the quality of the manuscript.

Reviewer 2 Report

Comments and Suggestions for Authors

I am satisfied with your answers to my comments.

Comments on the Quality of English Language

The article is written in a way that allows a good understanding of the text.

Author Response

We sincerely appreciate the reviewer's insightful comments, which have significantly enhanced the quality of the manuscript.